# Evaluation of Neuromuscular Fatigue in a Repeat Sprint Ability, Countermovement Jump and Hamstring Test in Elite Female Soccer Players

**DOI:** 10.3390/ijerph192215069

**Published:** 2022-11-16

**Authors:** Estrella Armada-Cortés, José A. Benítez-Muñoz, Alejandro F. San Juan, Javier Sánchez-Sánchez

**Affiliations:** 1Sport Biomechanics Laboratory, Department of Health and Human Performance, Faculty of Physical Activity and Sport Sciences-INEF, Universidad Politécnica de Madrid, C/Martín Fierro 7, CP, 28040 Madrid, Spain; 2LFE Research Group, Department of Health and Human Performance, Faculty of Physical Activity and Sport Sciences-INEF, Universidad Politécnica de Madrid, 28040 Madrid, Spain; 3School of Sport Sciences, Universidad Europea de Madrid, 28670 Villaviciosa de Odón, Spain

**Keywords:** fatigue, female soccer, sport, injury risk

## Abstract

The straight-line run is the most frequent action in soccer goal scoring situations, and it deserves considerable attention. The objective of this study was to evaluate the neuromuscular fatigue produced by an independent repeat sprint ability (RSA) test, a countermovement jump (CMJ) and a hamstring test (HT) in elite female soccer players. Twenty-four elite female soccer players participated in the study. The evaluation protocol included hamstring and CMJ tests before an RSA test (6 × 40 m 30 s rest), and hamstring and CMJ post-tests. Significant differences were found between pre–post HT measurements in the maximum angulation of the right leg (*p* = 0.012 Effect Size (ES) = 0.27), and the maximum velocity was higher in the left leg after RSA (*p* = 0.023 ES = 0.34). CMJ height after RSA was significantly lower than before the RSA test (*p* < 0.001 ES = 0.40). The sprint total time (SprintTT) and percentage difference (%Dif) increased throughout the RSA (*p* < 0.001, and ES = 0.648 and ES = 0.515, respectively). In elite female soccer players, it seems that the fatigue induced by an RSA test can be assessed through the loss of CMJ height and the different performance variables extracted from the RSA itself (e.g., Sprint_TT_, Ideal Sprint). These findings could contribute to better performance management and injury prevention for elite female soccer players.

## 1. Introduction

There are several physical demands required by soccer as a sport; these include: endurance, acceleration, deceleration, jumping, maximal sprinting, and repeated sprinting ability (RSA) [1]. Sprinting is a fundamental component of the professional soccer player’s ability to win duels, defend, or create scoring chances [2]. The straight-line run is the most frequent action in goal situations (i.e., both for the player who scores, and who assists) [3]. Consequently, the development and monitoring of repeated sprinting and sprinting abilities in soccer players deserve considerable attention.

Hamstring muscles are very important in sprint acceleration performance and maximal sprinting [4]. Fatigue during the later stages of the soccer match may cause an increased predisposition to hamstring strain injury by negatively altering the biomechanics of sprinting in relation to muscle flexibility, muscular strength, or body mechanics [5]. Therefore, most studies examining the mechanical and metabolic responses to repeated sprints used protocols with a fixed number of sprints and recovery periods of less than one minute to generate the fatigue and responses that are similar to game play [6,7].

It is generally accepted that strength and flexibility are important performance variables that have bearing upon the propensity for hamstring strain injury [8,9,10]. Most hamstring strain injuries in soccer occur during running [11,12], most often at the end of the match or training session [13], thus, suggesting that neuromuscular fatigue plays an important role [10,14]. Therefore, there is a need for a specific test, with performance and/or clinical validity, that can identify those players who are at risk of hamstring strain injury.

Dynamic strength tests, in particular eccentric tests (e.g., hamstring test (HT)), seem to be more suitable than static tests [15]. To date, the main risk factors reported for hamstring strain injury in soccer are previous injuries [16,17,18], muscle power imbalance [15,17], neuromuscular disorders [19], and fatigue [5,20,21].

The quality and quantity of hip range of motion (ROM) could affect the lack of hamstring extensibility in functional tasks, such as running [22]. Dynamic tests of extensibility have been proposed to be more sensitive to remaining abnormalities and are better tools to decide when return to play [23].

On the other hand, one of the most used performance tests is the vertical jump (VJ). Among them, countermovement jump (CMJ) performance has been used to monitor (1) the positive effects of strength, endurance, speed, and plyometric training and (2) the state of mechanical and neuromuscular fatigue in both individual and team sports [24]. In this regard, several researchers have found CMJ performance as an interesting and objective marker of fatigue and overcompensation of athlete performance [24], as well as one of the factors related to the high incidence of injuries (i.e., muscle overload) in lower extremity muscles [14,21]. Thus, a relationship between CMJ height loss and metabolic markers (i.e., such as lactate or ammonium) has been observed in sprinting [25]. This suggests that through the decrease in CMJ mechanical variables (i.e., jump height (JH)), it should be possible to estimate muscle fatigue, metabolic stress, and athlete overload [25].

Therefore, the objective of this study was to analyze the influence of the neuromuscular fatigue produced by an RSA test in elite female soccer players on the physical performance and hamstring extensibility. We hypothesized that after the RSA, its own performance variables, the CMJ height, and the hamstring’s extensibility, will decrease. Its uses could contribute to better performance management and injury prevention for elite female soccer players.

## 2. Materials and Methods

### 2.1. Experimental Approach to the Problem

An RSA protocol was performed to induce fatigue in the subjects (6 × 40 m 30 s rest) [26,27]. In addition, before and after the RSA, the HT was performed with five ballistic hip flexions in both legs, as well as three CMJ tests, to evaluate the fatigue generated and its repercussion on performance in the different tasks and the RSA itself.

### 2.2. Subjects

Twenty-four elite female soccer players of the Second Spanish Soccer Division participated in the study (aged 20.41 ± 2.63 years, height 166.75 ± 6.63 cm, and weight 59.79 ± 8 kg). All players had medical clearance to conduct the study and were completely healthy and uninjured at the time of the data collection. The study data were taken in the 2020–2021 season. All players had the same training load (i.e., five training days per week and a regular league match). The inclusion criteria were: (1) having medical clearance to conduct the study; (2) not having suffered a musculoskeletal injury 1 year prior to the date of the protocol (i.e., checked through a previous exclusion questionnaire); and (3) not being diagnosed with any cardiovascular, metabolic, neurologic, pulmonary, or orthopedic disorder that could affect participation in the study or limit the performance in the different tests.

All the athletes signed the informed consent form before the study. The study followed the guidelines of the Declaration of Helsinki and was approved by the Ethics Committee of the Universidad Politécnica de Madrid (Madrid, Spain) (code: DATOS-20211222-1SJF-Diseño de).

### 2.3. Procedures

The study comprised a pilot familiarization test session with the test protocols (i.e., RSA, CMJ, and HT), and a single evaluation session was performed on the team’s own artificial turf training and playing field. We performed all testing sessions under similar environmental conditions (20–24 °C, 45–55% relative humidity). The weight was calculated by the Withings Body+ scale (Issy-les-Moulineaux, France). All the participants used their appropriate personal footwear and sports equipment (i.e., short-sleeved T-shirt, shorts, and training socks). The test session was on the same schedule to avoid detrimental performance effects associated with circadian rhythm [28]. To ensure similar nutritional intake (60% carbohydrates, 30% lipids, and 10% proteins) from 72 h before the study onset, until its end, participants were provided with a set of guidelines. All the measures were realized by expert researchers (i.e., graduated and experienced physical trainers with a MSc or a PhD in physical activity and sport sciences). The sessions began with a 10 min general warm up, always guided by the club’s physical trainer, consisting of continuous and specific running, joint mobility, and ballistic stretching exercises, followed by a specific pre-test warm-up where participants performed three progressive sprints with 30 s of rest [27] between each one. After 180 s of rest [27], the participants started the protocol consisting of a pre-RSA HT, a pre-RSA CMJ test, the RSA test (6 × 40 m 30 s), a post-RSA HT, and a post-RSA CMJ test (Figure 1).

#### 2.3.1. Hamstring Test

The HT followed the WIMU^®^ hamstring test guide protocol [29]. The participant was placed lying down in supine position. A WIMU^®^ system (RealTrack Systems, Almería, Spain) was placed over the distal tibia, strapped with an elastic band to measure the inclination. The system is a small, wireless device with more than 20 integrated sensors. The sensors include a 1000 Hz 3D accelerometer, a 1000 Hz 3D gyroscope with 2000 degrees per second resolution, a 3D magnetometer, and a barometer that works with an integration of sensors to improve the measurements. All data regarding the angle achieved in each repetition for both legs were sent in real time via Bluetooth to a personal computer and subsequently recorded using SPro software (RealTrack Systems, Almeria, Spain). Participants performed five ballistic hip flexions, maintaining extension of the knee. The pelvis and the contralateral leg were fixed by a researcher to avoid pelvic movement, as modifications in pelvic positions have been reported to be factors that could affect the test score [30]. The WIMU^®^ system simultaneously recorded the hip flexion angle. This HT protocol was performed on both legs equally. An average of the five repetitions before and after the RSA test of the maximal velocity (deg/s), maximal angulation (deg), angle at maximal velocity (deg), time to maximal velocity (ms), and average velocity (deg/s) of each leg separately were analyzed, as well as the average of both legs.

#### 2.3.2. CMJ Test

All participants were familiar and fully acquainted with the CMJ technique. Participants were always instructed to jump as high as possible. The initial position consisted of a static standing position with hands on hips. From this position, a continuous and fast triple flexion movement of the hip, knee, and ankle was performed until they reached ~90° of knee flexion, followed by the triple extension of the same joints in a fluid, fast, and continuous manner [31]. In this type of vertical jump, there was a stretching–shortening cycle that took place during the consecutive eccentric, isometric, and concentric phases [31]. Participants were asked to take off and land the jump in the same place to avoid horizontal or lateral displacement. During flight, the participants were instructed that they must keep their hands on their hips and their legs should remain straight, with the ankle as extended as possible and contacting the ground with the tips of their feet. The aim was to reach the maximum height mean of the three jumps accomplished [32]. The jumps were evaluated through an Optojump photocell system (Microgate, Bolzano, Italy), which consists of two parallel bars (one receiver and one transmitter unit) that are positioned at the floor level, allowing for the athlete–surface interaction to be respected [33]. The Optojump photocell system was used extensively for field-based assessments and for research purposes [13,34,35], having been assessed for validity and reliability [33].

#### 2.3.3. RSA Test

The players were positioned 0.5 behind the first pairs of photocells, at the marked start of the sprints, to facilitate the correct registration of the first cut of the photocells [36]. The RSA test consisted of six repetitions of a 40 m flat sprint with 30 s rest in between each one. 

This distance allows for the athlete to reach their maximum speed [2]. The entire sprint course was monitored with a system of five pairs of photocells (Microgate, Bolzano, Italy) placed along the sprint zone to record the sprint time at 10, 20, 30, and 40 m with a sensibility of 0.001 s (Figure 2). The remaining recordings were taken in the same place that the athletes started and finished the sprint, so that the sprints were run back and forth. Different variables were calculated to evaluate fatigue during the RSA test (see the variables description in Table 1).

### 2.4. Statistical Analysis

Statistical power of the sample size was calculated with the software G power (Version 3.1.9.7, Stuttgart, Germany). It was performed a post hoc analysis for the ANOVA: repeated measures within factors taking as a reference the variable “Sprint_TT_ 0–40 m through the six sprints” with the following input parameters: effect size F: 1.73; α: 0.05; sample size: 24; groups: 1; measurements: 5, correlation: 0.5. According to these criteria, the power of the sample size was 1. All the data are presented as means ± SD. Normal distribution was confirmed using the Shapiro–Wilk test. Paired student t test was carried out to compare the different variables of the HT between right and left legs before the RSA, and pre–post measurements of the HT variables and CMJ height. Effect size for two means comparisons was calculated by Cohen’s d considering d < 0.5 as small, d < 0.8 as moderate, and d > 0.8 as large [38]. A one-way repeated measures ANOVA was used to analyze the different variables of the RSA test. When the Mauchly sphericity assumption was not met, the Greenhouse–Geisser correction was used. Bonferroni post hoc tests were conducted where significant differences were found in any of the analyzed factors. The effect size of the ANOVA was calculated by partial eta-squared (ηp^2^) and a small, moderate, and large effect corresponded to values equal or greater than 0.001, 0.059, and 0.138, respectively [39]. Data were analyzed using the SPSS statistic software, version 25.0, for Windows (IBM Corporation Armonk, NY, USA). The significance level was set at *p* < 0.05.

## 3. Results

### 3.1. Hamstring Test

Regarding pre–post comparisons, there was a significant difference between the pre–post measurements in the maximal angulation of the right leg (t (24) = 2.4 *p* = 0.012 ES = 0.27). Contrary, the maximal angulation of the right leg or the average angulation of both legs was not significantly different (*p* > 0.05). The maximal velocity of the left leg was higher after the RSA in comparison to before (t (24) = 2.1 *p* = 0.023 ES = 0.34). In contrast, there was no significant difference between the pre–post measurements in the maximal velocity of the right leg or the average of both legs (*p* > 0.05). Average velocity, time to maximal velocity, and angle at maximal velocity of the right leg, left leg, and the average of both legs was similar before and after the RSA (*p* > 0.05).

With respect to comparisons between legs, a significant difference was not found in the maximal angulation of the right leg compared to the left leg before the RSA (*p* > 0.05). On the contrary, a lower maximal angulation of the right leg compared to the left leg after the RSA was found (t (24) = 2.43 *p* = 0.023 ES = 0.25). The maximal velocity of the right leg was higher compared to the left leg before the RSA (t (24) = 3.76 *p* = 0.001 ES = 0.83) and after the RSA (t (24) = 2.49 *p* = 0.02 ES = 0.51). Average velocity of the right leg was higher compared to the left leg before the RSA (t (24) = 2.85 *p* = 0.009 ES = 0.42) and tended to be higher after the RSA (t (24) = 1.72 *p* = 0.099 ES = 0.36). The time to maximal velocity of the right leg was lower compared to the left leg before the RSA (t (24) = 2.52 *p* = 0.019 ES = 0.46) and tended to be lower after the RSA (t (24) = 1.96 *p* = 0.062 ES = 0.48). Finally, the angle at maximal velocity was not significantly different between legs before the RSA (*p* > 0.05), but it tended to be lower in the right leg compared to the left leg after the RSA (t (24) = 2.02 *p* = 0.055 ES = 0.48) (Table 2).

### 3.2. CMJ Test

Post-exercise CMJ height was significantly lower than before the RSA test (t (23) = 5.37 *p* < 0.001 ES = 0.40) (Table 2).

### 3.3. RSA Test

During the RSA, some distances have been registered showing the following results, increasing through the six sprints (Table 3): Sprint_TT_ 0–10 m (F (3.69) = 8.27 *p* < 0.001 ES = 0.256), Sprint_TT_ 0–20 m (F (2.49) = 10.47 *p* < 0.001 ES = 0.304), Sprint_TT_ 0–30 m (F (2.50) = 22.41 *p* < 0.001 ES = 0.483), Sprint_TT_ 0–40 m (F (2.49) = 44.17 *p* < 0.001 ES = 0.648). The %Dif between Sprint 1 compared to the rest of the sprints increased during the RSA (F (2.40) = 22.28 *p* < 0.001 ES = 0.515). The %Dif between Sprint_TT_ 0–40 m and Ideal Sprint became larger as the RSA progressed (F (1.27) = 62.50 *p* < 0.001 ES = 0.749) (Table 3). 

Figure 3 illustrates how both the Split Total Time (Split_TT_) (F (2.58) = 3780.33 *p* < 0.001 ES = 0.994) and Ideal Split (F (1.26) = 6535.72 *p* < 0.001 ES = 0.996) increase through the RSA. Higher Split_TT_ 0–10 m compared to Ideal Split 0–10 m (t (24) = 5.2 *p* < 0.001 ES = 0.82), Split_TT_ 0–20 m compared to Ideal Split 0–20 m (t (24) = 7.05 *p* < 0.001 ES = 0.83), Split_TT_ 0–30 m compared to Ideal Split 0–30 m (t (24) = 9.95 *p* < 0.001 ES = 0.80), and Split_TT_ 0–40 m compared to Ideal Split 0–40 m (t (24) = 6.74 *p* < 0.001 ES = 0.59) were found (Figure 3).

The percentage difference between the best and worst (%DifBvsW) Split_TT_ was significantly different between different distances of the RSA (F (1,33) = 5.06 *p* = 0.022 ES = 0.174) (Figure 4). On the contrary, the %Dif between Split_TT_ and Ideal Split was not significantly different between different distances of the RSA (F (3.72) = 0.89 *p* = 0.448 ES = 0.036) (Figure 4).

## 4. Discussion

The objective of this study was to analyze the influence of the neuromuscular fatigue produced by an RSA test in elite female soccer players on physical performance and hamstring extensibility. For this purpose, the incidence of neuromuscular fatigue was determined through a CMJ test, HT, and the performance variables of the RSA test itself. The primary results of the present study show that neuromuscular fatigue could be evaluated using the CMJ height, and the performance variables of the RSA itself. However, most of the HT variables were not affected by fatigue.

Some authors argue that the jump result average is more sensitive than the highest jump in detecting fatigue or supercompensation effects [24], thus, supporting the methodology of this study. We have found that the post-test CMJ height was significantly lower than the pre-test height (Table 3), agreeing with previous authors after a sprint test [25] or cycling test [40]. The decreasing response of jump height through the influence of fatigue has been seen in numerous studies [24,40,41,42]. In addition, these results have been extrapolated to different sport modalities such as soccer [40,41], team ball sports [42], and boxing [43]. Some authors highlight soccer and rugby as ball-implement sports where the recovery time for sprint performance is longer compared to other ball sports [42], therefore, it seems important to be able to quantify variables that give information about the fatigue state of the players.

In this way, a relationship has been observed between metabolic markers (such as lactate or ammonium) and the loss of jump height in the sprint task. This suggests that through decreases in CMJ height, it should be possible to estimate the metabolic stress, neuromuscular fatigue, and overload of the athlete [25].

Regarding the HT, we chose a ballistic test because it represents the eccentric phase of the hamstring muscles, which occurs in the swing phase of the sprint [4]. In addition, several authors conclude that the risk of hamstring muscle injury increases in the last minutes of the match, induced by fatigue, and related to the eccentric work phase of the hamstring muscles [14,21]. In this sense, muscle tension in the hamstring musculature increases with fatigue and decreases hamstring flexion values [14,21]. Therefore, it would be appropriate to think that hamstring flexibility will decrease after inducing fatigue in the RSA. Therefore, the overall interpretation of the test suggests that RSA-induced fatigue did not affect hamstring flexibility, differing from the expectation based on other studies [14,21]. The values of maximal angulation of the HT observed in this study are very similar to previous studies performing an active test [4], or a passive test [23,30,44,45]. Nevertheless, such comparisons should be taken with caution because the HT of the present study was a ballistic test.

On the other hand, focusing on the comparisons between legs, it was already foreseen in the previous results discussed that differences in maximum angulation and maximum speed between legs would make sense. It is noteworthy that other variables such as mean velocity or time to maximal velocity showed differences between legs in the Hamstring pre-test measurement, as these were not reflected between before and after the HT. 

The effect of fatigue on performance is demonstrated through the different variables of the RSA. As expected, the Sprint_TT_ increased during the RSA. This is the common method for evaluating sprint performance [36]. It is also observed that the Sprint_TT_ of the different distances (i.e., Sprint_TT_ 0−10 m, Sprint_TT_ 0−20 m, Sprint_TT_ 0−30 m, and Sprint_TT_ 0−40 m) increased during the RSA test, indicating that the time to cover different distances of the sprint increased. Our results agree with a previous study in which 7 × 30 m sprints with 25 s recovery in between were performed, and this upward trend in Sprint_TT_ at different distances was observed (0−5 m, 0−10 m, and 0−30 m) [36]. However, taking the first sprint 0−10 m as reference, they found significant differences from the fourth sprint onwards, and in the 0−30 m sprint, from the second sprint onwards. In our study, taking the first sprint as reference, we found significant differences in all the sprint distances. This indicates that fatigue occurs earlier in our study. The increased percentage difference between the first sprint and another specific sprint (%Dif1vsX) represents the magnitude of fatigue, as the greater the %Dif1vsX, the larger the effect of fatigue. In another study, the percentage of the difference between each sprint with the previous one was compared; differences were found in the first four of seven sprints [36]. Although the methodology to evaluate fatigue differs with the present study, the results of both studies align.

Different variables are provided to evaluate inter-sprint fatigue (Sprint_TT_, Ideal Sprint, %Dif1vsX, and %Dif between Sprint_TT_ 0−40 m and Ideal Sprint), and intra-sprint fatigue (Split_TT_, Ideal Split, %DifBvsW, and %Dif between Split_TT_ and Ideal Split). Knowing the origin of fatigue, it is possible to design specific types of training to counteract its effects, increase performance, and reduce the risks of injury, since fatigue is highly related to injury [5,46]. For example, in Figure 4, you can see how fatigue was higher in the first 10 m of the sprint, during the acceleration phase in our specific and limited population. The deeper approach in the RSA variables accomplished in this study may allow for trainers and researchers to analyze the RSA test from new perspectives, to discover new strengths or weakness of their athletes, and to train specifically to improve them. 

## 5. Conclusions

In elite female soccer players, it seems that the fatigue induced by an RSA test can be assessed through the loss of CMJ height, and different performance variables extracted from the RSA itself. In contrast, the lack of significant differences in the HT suggests that it is not an accurate way to assess fatigue after an RSA. Furthermore, the lack of differences in the HT in a fatigue situation (i.e., after an RSA test) might indicate that hamstring extensibility is not affected by fatigue. In order to strengthen these conclusions and generalize them to other athlete populations, future research is needed with a bigger sample, using a more demanding soccer task in the HT to compare the RSA’s influence on neuromuscular fatigue in male versus elite female soccer players, as well as in other competition levels and sports.

## 6. Practical Applications

Jump height loss can be used as a measure of fatigue after an RSA. This could be a useful tool in most team sports, where running is a determining factor, to identify fatigue and the possible increased risk of injury. On the other hand, the different RSA variables described will give coaches information on how fatigue affects the athlete during repeated sprinting and lend the trainer the opportunity to improve the specific weaknesses detected. So far, there is little information on these protocols in elite female soccer players, so coaches will have reference data on female soccer players and can make decisions based on them.

## Figures and Tables

**Figure 1 ijerph-19-15069-f001:**
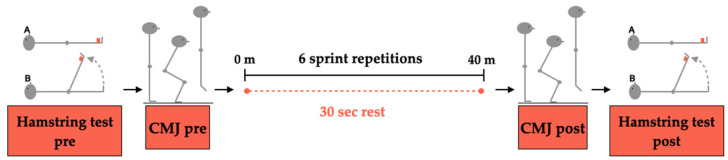
Outline of the protocol procedure. A: Initial position; B: Final position.

**Figure 2 ijerph-19-15069-f002:**
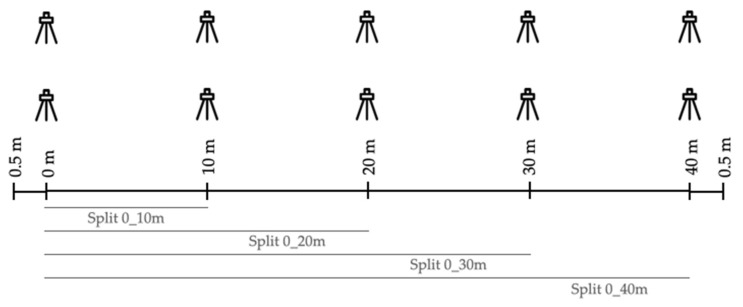
Diagram of the placement of the photocells in the RSA.

**Figure 3 ijerph-19-15069-f003:**
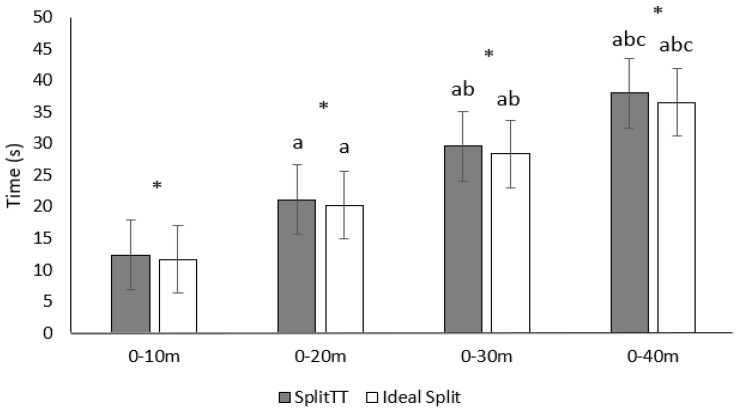
Evolution of the Split_TT_ and Ideal Split through the RSA. Split_TT_: the time spent to complete a specific distance during the six sprints, Ideal Split: the Split_TT_ if all the splits were run as the best of them. * Significant difference between Split_TT_ and Ideal Split. ^a^ Significant difference compared to 0−10 m (*p* < 0.05) ^b^ Significant difference compared to 0−20 m (*p* < 0.05) ^c^ Significant difference compared to 0−30 m (*p* < 0.05).

**Figure 4 ijerph-19-15069-f004:**
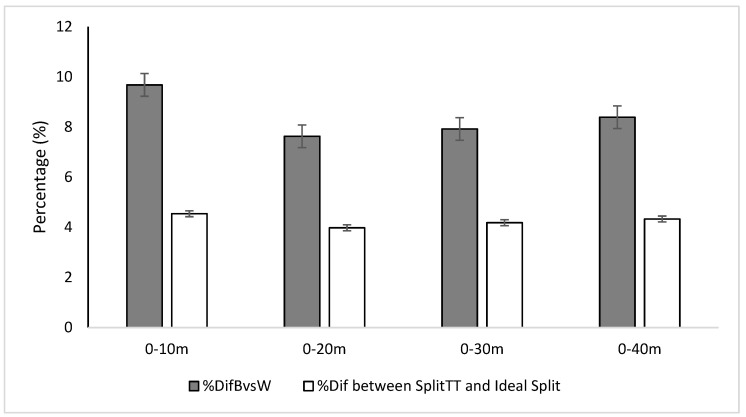
The %DifBvsW: Percentage difference between the best time compared to the worst time to run a specific split, %Dif between Split_TT_ and Ideal Split: Percentage difference between the time to run a specific number of splits and the time if these splits were run as the best of them. Significant difference compared to 0−10 m (*p* < 0.05).

**Table 1 ijerph-19-15069-t001:** Calculations of the different variables to evaluate fatigue during the RSA test.

Sprint Total Time (Sprint_TT_)	Defined as the Time to Run a Specific Distance of the Sprint	
Ideal Sprint [37]	Defined as the Sprint_TT_ if all the sprints were run as the best of them.	$\mathrm{Ideal}\;\mathrm{Sprint}\;1-X={\mathrm{MIN}\;\mathrm{Sprint}}_{\mathrm{TT}}0-40m\;\times 6$ Where X is the number of a specific sprint
Split Total Time (Split_TT_) [37]	Defined as the time spent to complete a specific distance during the six sprints.	${\mathrm{Split}}_{\mathrm{TT}}0-\mathrm{Xm}=0-\mathrm{Xm}\;\mathrm{Sprint}\;1+0-\mathrm{Xm}\;\mathrm{Sprint}\;2+\ldots +0-\mathrm{Xm}\;\mathrm{Sprint}\;6$ Where X is the distance of a specific split
Ideal Split [37]	Defined as the Split_TT_ if all the splits were run as the best of them.	$\mathrm{Ideal}\;\mathrm{Split}\;0-\mathrm{Xm}={\mathrm{MIN}\;\mathrm{Split}}_{\mathrm{TT}}\;0-\mathrm{Xm}\;\times 6$ Where X is the distance of a specific split
Percentage difference 1 vs. X (%Dif1vsX).	Defined as percentage difference between the first sprint and another specific sprint.	$\%\mathrm{Dif}=\frac{(({\mathrm{Sprint}}_{\mathrm{TT}}\;X-{\;\mathrm{Sprint}}_{\mathrm{TT}}\;1}{{\mathrm{Sprint}}_{\mathrm{TT}}\;1}\times 100$ Where X is the number of a specific sprint
Percentage difference between Sprint_TT_ 0–40 m and Ideal Sprint [27].	Defined as percentage difference between the time to run a specific number of sprints and the time if these sprints were run as the best of them.	${\%\mathrm{Dif}\;\mathrm{Sprint}}_{\mathrm{TT}}\;\mathrm{and}\;\mathrm{Ideal}\;\mathrm{Sprint}\;1-X=\frac{{\mathrm{Sprint}}_{\mathrm{TT}}\;X-\;\mathrm{Ideal}\;\mathrm{Sprint}}{\mathrm{Ideal}\;\mathrm{Sprint}}\times 100$ Where X is the number of a specific sprint
Percentage difference between the best time vs. worst time of a split (%DifBvsW) [27].	Percentage difference between the best time compared to the worst time to run a split.	$\%\mathrm{Dif}\;\mathrm{BvsW}\;0-\mathrm{Xm}=\frac{{\mathrm{MAX}\;\mathrm{Split}}_{\mathrm{TT}}\;0-\;\mathrm{Xm}-{\mathrm{MIN}\;\mathrm{Split}}_{\mathrm{TT}}\;0-\mathrm{Xm}}{{\mathrm{MIN}\;\mathrm{Split}}_{\mathrm{TT}}\;0-\mathrm{Xm}}\times 100$ Where X is the distance of a specific split
Percentage difference between Split_TT_ and Ideal Split (modified from [27]).	Percentage difference between the time to run a specific number of splits and the time if these splits were run as the best of them.	${\%\mathrm{Dif}\;\mathrm{Split}}_{\mathrm{TT}}\;\mathrm{and}\;\mathrm{Ideal}\;\mathrm{Split}\;0-\mathrm{Xm}=\frac{{\mathrm{Split}}_{\mathrm{TT}}\;0-\mathrm{Xm}-\mathrm{Ideal}\;\mathrm{Split}\;0-\mathrm{Xm}}{\mathrm{Ideal}\;\mathrm{Split}\;0-\mathrm{Xm}}\times 100$ Where X is the distance of a specific split

**Table 2 ijerph-19-15069-t002:** Results of the countermovement jump and hamstring test.

CMJ		PRE			POST	
Height (cm)		27.56 ± 5.09			25.56 ± 5.01 *	
**Hamstring test**	**R**	**L**	**Avg**	**R**	**L**	**Avg**
ROMmax (deg)	91.19 ± 13.7	89.2 ± 14.6	90.25 ± 13.43	87.55 ± 13.07 *	91.41 ± 17.12 ^#^	89.5 ± 14.7
Vmax (deg/s)	430.49 ± 98.82	365.9 ± 48.32 ^#^	398.36 ± 64.99	417.9 ± 76.86	383.83 ± 55.59 *^#^	406.64 ± 67.21
Vavg (deg/s)	256.91 ± 49.51	238.29 ± 39.13 ^#^	248.33 ± 42.14	264.52 ± 42.99	247.62 ± 51.38	256.07 ± 40.48
T to Vmax (ms)	98.13 ± 54.35	123.71 ± 57.84 ^#^	109.73 ± 49.72	86.7 ± 50.03	113.07 ± 59.11	106.14 ± 57.69
ROM at Vmax (deg)	38.79 ± 13.2	42.17 ± 14.63	40.50 ± 12.78	35.39 ± 13.59	42.29 ± 15.09	38.84 ± 11.55

Values presented as means ± SD. CMJ: countermovement jump, Max: maximal, Avg: average, Deg: degrees, R: right, L: left, ROMmax: maximal angulation, Vmax: maximal velocity, Vavg: average velocity, T to Vmax: time to maximal velocity, ROM at Vmax: angle at maximal velocity. * Significant difference between PRE compared to POST (*p* < 0.05). ^#^ Significant difference between legs (*p* < 0.05).

**Table 3 ijerph-19-15069-t003:** Performance variables of the repeated sprint ability.

	Sprint 1	Sprint 2	Sprint 3	Sprint 4	Sprint 5	Sprint 6
Sprint_TT_ 0−10 m	2 ± 0.12	2.05 ± 0.14 ^1^	2.03 ± 0.14	2.1 ± 0.14 ^1,2,3^	2.09 ± 0.14 ^1^	2.1 ± 0.14 ^1^
Sprint_TT_ 0−20 m	3.41 ± 0.17	3.5 ± 0.19 ^1^	3.48 ± 0.21 ^1^	3.57 ± 0.2 ^1,2,3^	3.57 ± 0.2 ^1,3^	3.58 ± 0.25 ^1^
Sprint_TT_ 0−30 m	4.76 ± 0.23	4.87 ± 0.25 ^1^	4.87 ± 0.29 ^1^	5 ± 0.29 ^1,2,3^	5 ± 0.29 ^1,2,3^	5.04 ± 0.31 ^1,2,3^
Sprint_TT_ 0−40 m	6.13 ± 0.31	6.29 ± 0.34 ^1^	6.31 ± 0.4 ^1^	6.47 ± 0.37 ^1,2,3^	6.5 ± 0.39 ^1,2,3^	6.6 ± 0.4 ^1,2,3,4^
		Sprint 1 vs. 2	Sprint 1 vs. 3	Sprint 1 vs. 4	Sprint 1 vs. 5	Sprint 1 vs. 6
%Dif		2.64 ± 1.47	3.05 ± 2.45	5.61 ± 2.58 ^a,b^	5.398 ± 3.64 ^a,b^	7.73 ± 4.7 ^a,b,d^
		**Sprint 1**−**2**	**Sprint 1**−**3**	**Sprint 1**−**4**	**Sprint 1**−**5**	**Sprint 1**−**6**
%Dif between Sprint_TT_ 0–40 m and Ideal Sprint		1.32 ± 0.74	2.03 ± 0.91 ^a^	2.96 ± 1.2 ^a,b^	3.59 ± 1.55 ^a,b,c^	4.33 ± 1.9 ^a,b,c,d^

Values presented as means ± SD. Sprint_TT_: Time to run a specific distance, %Dif: Percentage difference between the first sprint and a specific sprint, %Dif between Sprint_TT_ 0–40 m and Ideal Sprint: Percentage difference between the time to run a specific number of sprints and the time if these sprints were run as the best of them. ^1^ Significant difference compared to sprint 1 (*p* < 0.05). ^2^ Significant difference compared to sprint 2 (*p* < 0.05). ^3^ Significant difference compared to sprint 3 (*p* < 0.05). ^4^ Significant difference compared to sprint 4 (*p* < 0.05). ^a^ Significant difference compared to Sprint 1 vs. 2/Sprint 1−2 (*p* < 0.05). ^b^ Significant difference compared to Sprint 1 vs. 3/Sprint 1−3 (*p* < 0.05). ^c^ Significant difference compared to Sprint 1 vs. 4/Sprint 1−4 (*p* < 0.05). ^d^ Significant difference compared to Sprint 1 vs. 5/Sprint 1−5 (*p* < 0.05).

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
