# Peer review of "Evaluation of Neuromuscular Fatigue in a Repeat Sprint Ability, Countermovement Jump and Hamstring Test in Elite Female Soccer Players"

_ijerph, 2022, doi:10.3390/ijerph192215069_

Round 1
Reviewer 1 Report
Thank you for the opportunity to review this paper. The topic is of interest in its current form. Some areas require rewriting or clarification.
The rationale behind the study and objectives are presented.
How was the sample size determined?
Provide the inclusion criteria for participants.
Company name and model number should be provided for the stabilometer and weighing scale.
Do lines 125-165 have some meaning? Otherwise, remove them.
Provide the statistical analysis of data in detail. Does the data were checked for normality? Why the paired t-test was used?
Provide the table for vertical jump test results,
Provide day one and day two comparative results tables for peak cardiorespiratory variables.
Author Response
About this cover letter
In this cover letter we will respond to reviewers´ 1 comments one by one in blue. Please keep in mind that line numbers have changed so you should follow the corrected manuscript.
REVIEWER 1 ANNOTATIONS
Estimated reviewer,
Thank you for giving us the opportunity to improve our manuscript with your revision and useful suggestions. We hope that we have been able to adequately resolve all your comments.
Thank you again for your time and kind regards,
On behalf of all the authors of the manuscript.
"How was the sample size determined?"
Thank you for the comment. The statistical power calculation has been added correctly in the statistical analysis section
"Provide the inclusion criteria for participants"
Thank you very much for your appreciation but the inclusion criteria are set out in the text
"Do lines 125-165 have some meaning? Otherwise, remove them"
Thank you for your comment, the lines referred to contain important information on the procedure and methodology of the protocol. Therefore, we consider your explanation to be of vital importance for a possible later replication of the study
"Provide the statistical analysis of data in detail. Does the data were checked for normality? Why the paired t-test was used?"
Thank you for the comment. The text has been modified. Paired t-test was performed when two means were compared after confirming normal distribution. Such as in the case of comparing left-right or pre-post.
"Provide day one and day two comparative results tables for peak cardiorespiratory variables"
Thank you for your comment. In the study we do not have any cardiorespiratory data and then it is not possible to do comparisons between days one and two.
"Provide the table for vertical jump test results"
Thank you for the comment. Vertical jump results are in table 2. CMJ and hamstring test results are in the same table to avoid overloading with so many tables.

Reviewer 2 Report
Dear authors,
I would like to express my gratitude regarding the opportunity to review this manuscript.
The study topic is interesting, at this stage the manuscript requires considerable improvements. Below comments and suggestions with line indication:
2-4 – Please review the upper and lowercase in the title.
5 – Please change “&” for “and”.
19-35 – Please consider deleting the numbers “(1)” and topics “background” and others.
22 – Please review “and an over Countermovement Jump (CMJ), and a Hamstring Test”.
19-35 – Please consider changing the abstract text, namely the results and conclusions text.
36 – Please consider the journal template and instructions for authors in keywords.
39 – It seems more than one space after “:”, please review.
41 – RSA should be abbreviated in the first abbreviation in the manuscript.
59 – “Hamstring Test”, please consider and standardize upper and lowercase case. The same throughout the manuscript (abstract included). Please also consider abbreviating “Hamstring Test”, it appears too many times in the text in full.
59-62 and 63-76 – One paragraph is too short and the other too long, plase consider standardization.
78 – “CMJ” first appearance in the manuscript should be in full.
85 – “as lactate or ammonium, in sprinting [35].”- Please review the English.
96 – Please provide reference to support RAS methodology.
106 - It seems more than one space after “:”, please review.
116 - Please describe the procedures in detail. For example, where were the evaluations performed? Conditions (temperature, humidity?) Previous nutrition? Clothes during data collection? Who collected the data, training, and experience? All details should be considered and detailed.
125 – Please provide reference supporting the notion that rest between sprints and after warm-up was adequate. Also in this line, “s” and “minutes” – Please apply the same criteria in units throughout the manuscript.
128 – Please improve the figure quality. Also standardize upper and lowercase.
140 – Please consider paragraph.
154 – “SSC” not needed, “stretching-shortening cycle” only one time in the manuscript.
164 – Please change “( )” for “[ ]”.
177 – Please change figure type of letter and size according to the journal template and instructions for authors.
173-176 – It is suggested that this introductory text of table 1 is presented between figure 2 and table 1.
179 – Please review table 1 content.
181-193 – There are two indications related to ES. Please explain in detail. It is also important to describe the power of the sample size.
195-231 – This part of the results section should be carefully reviewed; it is suggested to be considered the text reformulation. Tables should be presented near the text describing the results. The tables should be formated aiming fast interpretation by readers.
232 – Please consider “ideal split” abbreviation throughout the manuscript.
238 – Please improve figure quality, according to the journal template and instructions for authors. Also space between values and “m” is suggested (similar to text and figure 2). Please also consider the best place in the page to present the figure (it is too much in the right side and above page line in V1). The same details should be considered in figure 4.
250 - It seems more than one space after “:”, please review.
268-285 – Paragraph is too long, please consider splitting to improved reading conditions.
284-285 – This sentence should be presented in the end of the discussion section, near the indication of study limitations.
309-329 - Paragraph is too long, please consider splitting to improved reading conditions.
329 - Please develop the limitations of the study and include suggestions for future research.
330 – Please consider highlighting the main findings of the study associated to take-home messages, and desirably, with practical applications.
350 – Please include space.
357 – Please review “This research was funded by the editorial.”.
358 – Not according to the journal template and instructions for authors. Please review.
363 – All references format should be corrected considering the journal template and instructions for authors.
Please carefully review all manuscript and consider English improvement.
Author Response
About this cover letter
In this cover letter we will respond to reviewers´ 2 comments one by one in blue. Please keep in mind that line numbers have changed so you should follow the corrected manuscript.
REVIEWER 2 ANNOTATIONS
Estimated reviewer,
Thank you for giving us the opportunity to improve our manuscript with your revision and useful suggestions. We hope that we have been able to adequately resolve all your comments.
Thank you again for your time and kind regards,
On behalf of all the authors of the manuscript.
"2-4 – Please review the upper and lowercase in the title"
Thank you very much for your comment. The capitalization of the text in question has been changed
"5 – Please change “&” for “and”"
Thank you for the suggestion, the text has been changed
"19-35 – Please consider deleting the numbers “(1)” and topics “background” and others"
Thank you for your comment, the numbers and topics been deleted
"19-35 – Please consider changing the abstract text, namely the results and conclusions text"
Thank you for the suggestion. The resulsts have been simplified and the conclusions modified
"22 – Please review “and an over Countermovement Jump (CMJ), and a Hamstring Test”"
Thank you for the suggestions, the text has been modified
"36 – Please consider the journal template and instructions for authors in keywords"
Thank you for the suggestion, other keyword has been added following the instruccions for authors
"39 – It seems more than one space after “:”, please review"
Thank you for you comment. The error has been modified
"41 – RSA should be abbreviated in the first abbreviation in the manuscript"
Thank you for the suggestion, it has been modified
"59 – “Hamstring Test”, please consider and standardize upper and lowercase case. The same throughout the manuscript (abstract included). Please also consider abbreviating “Hamstring Test”, it appears too many times in the text in full"
Thank you for the comment. Capitalization and abbreviations have been changed
"59-62 and 63-76 – One paragraph is too short and the other too long, plase consider standardization"
Thank you very much for your comment. The content of the paragraph has been modified and considerably reduced
"78 – “CMJ” first appearance in the manuscript should be in full"
Thank you for your comment. The text has been added
"85 – “as lactate or ammonium, in sprinting [35].”- Please review the English"
Thank you for the suggestion, the text has been modified
"96 – Please provide reference to support RAS methodology"
Thank you for the cooment. The references has been added
"106 - It seems more than one space after “:”, please review"
Thank you for the comment. It has been solved
"116 - Please describe the procedures in detail. For example, where were the evaluations performed? Conditions (temperature, humidity?) Previous nutrition? Clothes during data collection? Who collected the data, training, and experience? All details should be considered and detailed"
Thank you for the comment. The required information has been added
"125 – Please provide reference supporting the notion that rest between sprints and after warm-up was adequate. Also in this line, “s” and “minutes” – Please apply the same criteria in units throughout the manuscript"
Thank you for your comment. The unit of measurement has been changed and the reference added
"128 – Please improve the figure quality. Also standardize upper and lowercase"
Thank you for the comment, the quality and changes in the image have been modified
"140 – Please consider paragraph"
Thank you for your comment. This paragraph is necessary and explanatory about the methodology followed in the hamstring test proceidure
"154 – “SSC” not needed, “stretching-shortening cycle” only one time in the manuscript"
Thank you for the comment, the abbreviation has been removed
"164 – Please change “( )” for “[ ]”"
Thank you for the comment. The error has been solved
"177 – Please change figure type of letter and size according to the journal template and instructions for authors"
Thank you for the comment. The letter and size has been modified based on journal template and instruction for authors
"173-176 – It is suggested that this introductory text of table 1 is presented between figure 2 and table 1"
Thank you, the text place has been modified
"179 – Please review table 1 content"
Thank you for your comment. Table 1 is necessary to clearly describe the variables used in the repeated sprint test
"181-193 – There are two indications related to ES. Please explain in detail. It is also important to describe the power of the sample size"
Thank you for the comment. We have specified the ES. Also, the statistical power is shown
"195-231 – This part of the results section should be carefully reviewed; it is suggested to be considered the text reformulation. Tables should be presented near the text describing the results. The tables should be formated aiming fast interpretation by readers"
Thank you for the suggestion, The text has been reformulated
"232 – Please consider “ideal split” abbreviation throughout the manuscript"
Thank you for the cooment. This variable is defined as the Sprint total time if all the sprints were run as the best of them and it is very important for the understanding of the results
"238 – Please improve figure quality, according to the journal template and instructions for authors. Also space between values and “m” is suggested (similar to text and figure 2). Please also consider the best place in the page to present the figure (it is too much in the right side and above page line in V1). The same details should be considered in figure 4"
Thank you for the comment. The figure has been modified
"250 - It seems more than one space after “:”, please review"
Thank you for the comment, the error has been solved
"268-285 – Paragraph is too long, please consider splitting to improved reading conditions"
Thank you for the suggestion. The text has been modified
"309-329 - Paragraph is too long, please consider splitting to improved reading conditions"
Thank you for the comment. The text has been modified
"329 - Please develop the limitations of the study and include suggestions for future research"
Thank you for the suggestion, it has been written in the conclusions
"330 – Please consider highlighting the main findings of the study associated to take-home messages, and desirably, with practical applications"
Thank you for your comment, this has been explained in section "6. Practical applications".
"284-285 – This sentence should be presented in the end of the discussion section, near the indication of study limitations"
Thank you for the comment. The place of the sentence has been change
"350 – Please include space"
Thank you for the suggestion, the space has been added
"358 – Not according to the journal template and instructions for authors. Please review"
Thank you for the suggestion, it has been removed
"357 – Please review “This research was funded by the editorial.”"
Thank you for the comment. It has been modified
"363 – All references format should be corrected considering the journal template and instructions for authors"
Thank you for the comment. The references format has been modified

Round 2
Reviewer 2 Report
Dear authors,
Thank you for considering my suggestions and incorporating them into the manuscript.
Below suggestions related to this last version (v2), with line or page indication.
2,3,4 – Please review the title considering upper and lowercase and assuming IJERPH instructions for authors (https://www.mdpi.com/journal/ijerph).
24 and 86 – The format is different, (6x40m 30s rest / 6 x 40 m 30 s rest), please standardize. Considering the text in line 92, spaces should be assumed, and abstract correct. Please review.
111 – Please include instrument city and country.
121 – I think the citation should be presented after “rest”, please review.
161 – Please remove the space between citation numbers.
177 – It is once again suggested to carefully review the content of table 1. For example, a space should be presents before “Percentage difference between SplitTT and ideal Split (modified from [42]).”
185, 186 – Please correct to HT.
192-194 – Please provide a reference regarding “Effect size of the ANOVA was calculated by partial eta-squared”.
223 – Please consider better organizing the text in table 2 to provide readers with the best conditions for interpretation. Also consider reformulating the legend text for readers fast interpretation.
224-231 / 232-237 / 244-248– Please correct all values to “.” instead of “,”.
231 – Please correct all values to “.” instead of “,” and place all values with the same decimals.
239-240 – Please confirm if no more than one space after “:”.
Figure 2 presents “Ideal Split” and Figure 3 in lowercase (in the same page – 8). Please standardize.
254 – The discussion section only present one new citation in the text. It is possible to improve the discussion section.
341 until the end of manuscript text – The text format is not according to the journal instructions for authors (including the references), namely the paragraph size. Please correct.
347 – All refs need to be carefully reviewed and corrected. For example, the journals should be abbreviated; DOI´s should be provided; Ref 5 “…” corrected; between authors “;” and not “,”; ref 10 does not present volume, number, pages and DOI; and many other examples. This is extremely important and should be reviewed and corrected in detail.
Please carefully review the manuscript without track changes before providing V3. It is suggested that in the next review round, the manuscript is presented without track changes and in the cover letter the improvements and answers to reviewers be presented with line indication. Thank you.
Author Response
About this cover letter
In this cover letter we will respond to reviewers´ 2 comments, version 2, one by one in blue. Please keep in mind that line numbers have changed so you should follow the corrected manuscript.
REVIEWER 2 ANNOTATIONS
Estimated reviewer,
Thank you for giving us the opportunity to improve our manuscript with your revision and useful suggestions again. We hope that we have been able to adequately resolve all your comments.
Thank you again for your time and kind regards,
On behalf of all the authors of the manuscript.
"2,3,4 – Please review the title considering upper and lowercase and assuming IJERPH instructions for authors (https://www.mdpi.com/journal/ijerph)"
LINE 2, 3, 4 - Thank you for your comment. The capital letters have been modified based on the journal's guidelines in the hope that they are correct
"24 and 86 – The format is different, (6x40m 30s rest / 6 x 40 m 30 s rest), please standardize. Considering the text in line 92, spaces should be assumed, and abstract correct. Please review"
LINE 21 - Thank you for the suggestion, it has been standarized preserving the spaces
111 – Please include instrument city and country.
LINE 107 - Thank you for the suggestion, it has been added
121 – I think the citation should be presented after “rest”, please review.
LINE 118 - Thank you for the suggestion, the reference has been changed after "rest"
161 – Please remove the space between citation numbers.
LINE 157 - Thank you for the suggestion, it has been changed
177 – It is once again suggested to carefully review the content of table 1. For example, a space should be presents before “Percentage difference between SplitTT and ideal Split (modified from [42]).”
LINE 172 - Thank you for the suggestion. The content has been revised and modified
185, 186 – Please correct to HT
LINE 181, 182 - Thank you for the suggestion, it has been modified
192-194 – Please provide a reference regarding “Effect size of the ANOVA was calculated by partial eta-squared”.
LINE 190 - Thank you for the suggestion, the reference has been added
223 – Please consider better organizing the text in table 2 to provide readers with the best conditions for interpretation. Also consider reformulating the legend text for readers fast interpretation.
LINE 118 - Thanks for the comment, legends have been added for a better reading and interpretation of the results
224-231 / 232-237 / 244-248– Please correct all values to “.” instead of “,”.
LINE 221-226 - Thank you for the suggestion, it has been modified
231 – Please correct all values to “.” instead of “,” and place all values with the same decimals.
LINE 227-233 - Thank you for the suggestion, it has been modified
239-240 – Please confirm if no more than one space after “:”
LINE 236 - Thank you for the comment. One space has been deleted
Figure 2 presents “Ideal Split” and Figure 3 in lowercase (in the same page – 8). Please standardize
LINE 235 - Thank you for your comment. We understand that the modification was in figures 3 and 4 and based on them have been standardized
254 – The discussion section only present one new citation in the text. It is possible to improve the discussion section
LINE 282-285 - Thank you very much for the suggestion. The text of the discussion has been extended and increased in references
341 until the end of manuscript text – The text format is not according to the journal instructions for authors (including the references), namely the paragraph size. Please correct.
LINE 355-unitil the final of the text - Thank you for the suggestion, the format has been modified until the final of the text
347 – All refs need to be carefully reviewed and corrected. For example, the journals should be abbreviated; DOI´s should be provided; Ref 5 “…” corrected; between authors “;” and not “,”; ref 10 does not present volume, number, pages and DOI; and many other examples. This is extremely important and should be reviewed and corrected in detail
Thank you for the suggestion. All references have been reviewed and changed if is necessary
